# Awareness and Perspectives among Asian Anesthesiologists on Postoperative Delirium: A Multinational Survey

**DOI:** 10.3390/jcm10245769

**Published:** 2021-12-09

**Authors:** Hyungmook Lee, Jeongmin Kim, Ki-Young Lee, Tong J. Gan, Varinee Lekprasert, Prok Laosuwan, Sophia Tsong Huey Chew, Edwin Seet, Vera Lim, Lian Kah Ti

**Affiliations:** 1Department of Anesthesiology and Pain Medicine, Seoul St. Mary’s Hospital, College of Medicine, The Catholic University of Korea, Seoul 06591, Korea; warmy0828@catholic.ac.kr; 2Department of Anaesthesiology and Pain Medicine, Anaesthesia and Pain Research Institute, College of Medicine, Yonsei University, Seoul 03722, Korea; ANESJEONGMIN@yuhs.ac; 3Department of Anesthesiology, School of Medicine, Stony Brook University Renaissance, Stony Brook, NY 11794, USA; Tong.Gan@stonybrookmedicine.edu; 4Department of Anesthesiology, Faculty of Medicine, Ramathibodi Hospital, Mahidol University, Bangkok 10400, Thailand; varinee.lek@mahidol.ac.th; 5Department of Anesthesiology, Faculty of Medicine, Chulalongkorn University, King Chulalongkorn Memorial Hospital, Thai Red Cross Society, Bangkok 10330, Thailand; prokchula@gmail.com; 6Division of Anaesthesiology and Perioperative Medicine, Singapore General Hospital, Singapore 169608, Singapore; sophia.chew.t.h@singhealth.com.sg; 7Department of Anaesthesia, Khoo Teck Puat Hospital, Singapore 768828, Singapore; seet.edwin.cp@ktph.com.sg; 8Department of Anaesthesia, Tan Tock Seng Hospital, Singapore 308433, Singapore; vera_qy_lim@ttsh.com.sg; 9Department of Anaesthesia, National University Health System, Singapore 119228, Singapore; anatilk@nus.edu.sg

**Keywords:** anesthesiology, consciousness monitors, NIR spectroscopy, perioperative care, postoperative cognitive complications, postoperative complications

## Abstract

Postoperative delirium (POD) is a common perioperative complication. Although POD is preventable in up to 40% of patients, it is frequently overlooked. The objective of the survey is to determine the level of knowledge and clinical practices related to POD among anesthesiologists in different Asian countries. A questionnaire of 22 questions was designed by members of the Asian focus group for the study of POD, and it was sent to anesthesiologists in Singapore, Thailand, and South Korea from 1 April 2019 through 17 September 2019. In total, 531 anesthesiologists (Singapore: 224, Thailand: 124, Korea: 183) responded to the survey. Half the respondents estimated the incidence of POD to be 11–30% and believed that it typically occurs in the first 48 h after surgery. Among eight important postoperative complications, POD was ranked fifth. While 51.4% did not perform any test for POD, only 13.7% monitored the depth of anesthesia in all their patients. However, 83.8% preferred depth of anesthesia monitoring if they underwent surgery themselves. The results suggest that Asian anesthesiologists underestimate the incidence and relevance of POD. Because it increases perioperative mortality and morbidity, there is an urgent need to educate anesthesiologists regarding the recognition, prevention, detection, and management of POD.

## 1. Introduction

Of late, concerns regarding the association between surgery and/or anesthesia and perioperative neurocognitive disorder (NCD), including postoperative delirium (POD), have been increasing. Several factors such as repeated exposure to surgery and/or anesthesia, hypothermia during surgery, excessive neuroinflammation, and an excessive depth of anesthesia are associated with POD [1]. Conventionally, acute neurocognitive decline after surgery is termed POD, which is associated with significant postoperative morbidity and mortality. POD is also associated with prolonged hospitalization, delayed neurocognitive recovery, limitations in basic activities of daily living, and even mortality [2,3,4].

POD is a common perioperative complication, with the reported incidence being as low as 1.4% in an ambulatory setting and as high as 60% in patients undergoing emergency major vascular surgeries [5,6]. Furthermore, the Diagnostic and Statistical Manual of Mental Disorders (DSM-5) states that delirium occurs in 15–53% of older individuals who have undergone surgery and 70–87% of older individuals in intensive care [7]. POD is not limited to adults. Emergency delirium or agitation is usually more common in preschool children, with an incidence of 10–20% in many studies, but up to 66% has been reported [8,9]. However, POD is preventable in up to 40% of patients [10].

The aim of the survey was to determine the level of knowledge and clinical practices related to POD among anesthesiologists in different Asian countries and raise awareness regarding NCD in these regions.

## 2. Materials and Methods

### 2.1. Survey Development

This was an international multicenter survey. Specialist anesthesiologists, anesthesia residents, and junior doctors were invited to participate. The questionnaire for the survey (Supplementary Methods) was designed and developed by five members (two from Korea (K), two from Thailand (TH), and one from Singapore (SG)) of the Southeast and East Asian Post-Operative Delirium (SEAPOD) study group for the study of postoperative delirium.

The POD focus group members distributed the survey in SG, TH, and K from 1 April 2019 through 17 September 2019 (Figure 1). In TH, the questionnaire was privately distributed to several university and rural hospitals via direct electronic mails and social networking sites. In K, the questionnaire was sent to four tertiary teaching hospitals. In SG, the survey was sent to five public hospitals. Participants from K and SG received a printed questionnaire.

### 2.2. Questionnaire

The questionnaire included 22 questions that would require 5 to 10 min to complete. The first set of questions were related to the background characteristics of the respondents, including their age, sex, main location of practice, and duration of anesthesia practice in years. The second set of questions asked about knowledge and perspectives regarding POD, including the ranking of postoperative complications, typical time of occurrence, most common form, associated morbidities and mortality, frequency of occurrence, associated types of anesthesia, and risk factors. The third set of questions was related to practices concerning POD, including the frequency of tests, frequency of depth of anesthesia monitoring, reasons for depth of anesthesia monitoring, factors limiting the use of depth of anesthesia monitoring, effects of depth of anesthesia monitoring on POD, cost-effectiveness of depth of anesthesia monitoring, choice of depth of anesthesia monitoring for oneself, and use and usefulness of cerebral oximetry. Survey questions and responses are available on Mendeley Data (doi:10.17632/rc3jkwhf7f.2) (accessed on 9 December 2021).

A total of 1000 questionnaires were delivered to anesthetists in three countries. There were 531 total responses, and the overall response rate was 57.4%. In SG, the questionnaires were distributed to 300 anesthetists, and 224 of them completed the survey. The response rate was 74.7%. In TH, the questionnaires were distributed to 420 anesthetists, and 124 of them completed the survey. The response rate was 29.5%. In K, the questionnaires were distributed to 205 anesthetists, and 183 of them completed the survey. The response rate was 89.3%. The total number of anesthetists working is 550 (SG), 2328 (TH), and 3300 (K). Response rates compared to the total number of working anesthetists were 40.7% (SG), 5.3% (TH), and 5.5% (K)

The questionnaire was distributed in a manner that was expected to receive the most responses in each country. In SG, the survey was explained during regular department meetings in five major public health institutions in Singapore. Surveys were placed on the desk or given out face-to-face. In K, the author explained the purpose of the survey at a regular meeting with the heads of the departments of anesthesiology of major teaching hospitals. The questionnaire was sent to the four hospitals that wanted to participate in the survey. In TH, the survey questionnaire was distributed in an annual meeting of teaching hospitals. In addition, the surveys were sent to the two tertiary hospitals in Bangkok.

Postoperative neurocognitive decline is not well defined, with both POD and postoperative cognitive dysfunction (POCD) used interchangeably. In 2018, a group of experts published recommendations regarding the nomenclature for perioperative NCD in order to enhance the consistency of communication and reporting [11]. They defined POD as cognitive dysfunction occurring in the hospital up to 1 week after the procedure or until discharge (whichever is earlier) and meeting the DSM-5 diagnostic criteria. For POCD, they recommended the use of delayed neurocognitive recovery and NCD instead of POCD, which should be used only as a specifier. However, for the purpose of the current survey, we used POD and POCD interchangeably because the change in nomenclature has not been widely accepted by anesthesiologists in the regions covered by the survey.

### 2.3. Statistical Analysis

Descriptive statistics (absolute frequencies and percentages) for all responses and the results of one-way analysis of variance (ANOVA) for comparisons of responses among the three countries were analyzed using GraphPad Prism version 7.04 for Windows (GraphPad Software, La Jolla, CA, USA).

### 2.4. Ethics Statement

This study was conducted in accordance with the principles of Good Clinical Practice and was approved by the institutional review board in each of the three Asian countries. In TH, approval was obtained from the Faculty of Medicine, Ramathibodi Hospital, Mahidol University (Reference No. MURA2021/604). In K, approval was obtained from the Severance Hospital (Reference No. 4-2018-0833) and Seoul St. Mary’s Hospital (Reference No. KC18QEGI0819), while in SG, approval was obtained from the Domain Specific Review Board (DSRB Reference No. 2018/01154). Informed consent was obtained at the time of questionnaire administration from all participants in K, whereas a waiver of consent was sought and approved for all participants in SG and TH.

## 3. Results

### 3.1. Demographics of Study Participants

There were 531 responses, including 224 from SG, 124 from TH, and 183 from K. In total, 40.5% respondents were aged between 31 and 40 years, and the number of female respondents (55.7%) was slightly higher than that of male respondents (44.3%). While 63.1% respondents worked in a university hospital, 29% worked in a city hospital and 7.9% in a regional or rural hospital. The duration of anesthesia practice was evenly distributed as follows: <5 years, 34.7%; 5 to 10 years, 27.3%; 10 to 15 years, 22.2%; and >15 years, 15.8% (Table 1).

### 3.2. Knowledge and Perspectives Regarding POD

For elderly patients undergoing non-cardiac surgery, the frequency of POD occurrence was estimated as less than 10% by 18.3% respondents, 11% to 30% by 51.4% respondents, 31% to 50% by 22.0% respondents, and more than 50% by 8.3% respondents.

We asked participants about the importance of eight postoperative complications (1: most important, 8: least important). The anesthesiologists from all three countries were consistent in ranking myocardial infarction, cerebral stroke, and hypoventilation and hypercapnia as the three most important complications. Overall, POD was ranked fifth in importance.

Among perioperative cognitive states, intraoperative awareness was consistently ranked as the most important. POD and POCD were given less importance, and emergence agitation was considered the least important (Table 2).

### 3.3. Awareness of Patient and Anesthetic Risk Factors for POD

The majority of respondents answered that old age (98.1%) and underlying cognitive impairment (90.8%) were risk factors for POD, followed by frailty (76.5%), poor nutrition (76.1%), anemia (56.3%), renal insufficiency (52.7%), diabetes (48.6%), genetics (44.8%), low education level (39.9%), and gut microbiota (17.3%) (Table 3).

More than 96% of respondents were of the opinion that general anesthesia is associated with POD, whereas 30.5%, 69.7%, and 66.1% associated POD with regional anesthesia without sedation, regional anesthesia with sedation, and total intravenous anesthesia (TIVA), respectively (Figure 2).

With regard to the choice of drugs, the majority of respondents thought that midazolam (84.7%), inhalational agents (81.4%), ketamine (74.4%), opioids (71.9%), and anticholinergics (70.1%) contributed to POD, whereas 38.4% considered propofol to be a contributing factor. A small proportion of respondents associated dexmedetomidine (20%) and muscle relaxants (17.3%) with POD.

### 3.4. Perspectives on Testing and Monitoring

Approximately half the respondents (51.4%) did not perform any POD tests, with a few performing a test for all patients in the postanesthesia care unit (PACU; 5.6%) and general ward (3.2%).

More than three-quarters of the respondents (76.6%), including 66.1% from SG, 90.3% from TH, and 80.3% from K, thought that depth of anesthesia monitoring can prevent POD.

In total, 52% respondents used depth of anesthesia monitoring for POD prevention. Other reasons for depth of anesthesia monitoring included prevention of intraoperative awareness (88.9%), management of TIVA (67.2%), faster awakening (49.7%), prevention of mortality (23.4%), shortening of the PACU stay (22.8%), and prevention of postoperative nausea and vomiting (14.5%).

Overall, 13.7% of respondents used depth of anesthesia monitoring for all patients. Approximately a quarter (25.1%) of respondents from K and 13.7% and 8.9% respondents from TH and SG, respectively, reported this as routine practice.

The proportion of respondents who used depth of anesthesia monitoring in more than two-third of the total number of anesthesia cases, but did not perform it in routine practice, was 20.5%; 14.7% mentioned that they performed depth of anesthesia monitoring in <10% of cases. In TH, 18.5% of respondents did not perform monitoring because of unavailability, whereas unavailability was not mentioned by any respondent from SG and K.

The cost (47.5%) and availability of equipment (43.1%) were the main factors that limited routine use of depth of anesthesia monitoring in cases of general anesthesia, particularly in SG (50.4%) and TH (53.2%). In K, this rate was lower, at 39.9%.

Overall, 42% of respondents felt that depth of anesthesia monitoring was cost-effective only in high-risk cases, and 45.4% and 18.5% from K and TH, respectively, opined that it was cost-effective in the majority of cases. However, >80% of respondents (83.8%), including 95.1% from K, 87.9% from TH, and 72.3% from SG, themselves preferred to undergo surgery with depth of anesthesia monitoring.

With regard to the use of cerebral oximetry, approximately a third of the respondents used it on an infrequent basis, while 19.2% did not use it because of unavailability at their place of practice, and 12.6% did not use it even though it was available at their place of practice. Only 3.8% of respondents used it in >50% cases.

Cerebral oximetry was considered useful for the prevention of cerebral stroke by 77% of respondents, whereas 57.4% and 48.4% considered it useful for the prevention of POCD and POD, respectively. Approximately one-tenth of the respondents (9%; 10.7% from SG, 8.7% from K, and 6.5% from TH) thought it was not a useful monitoring tool.

### 3.5. Morbidity and Mortality Associated with POD

More than 80% of respondents associated POD with dementia (80.6%) and prolonged hospitalization (86.4%). Only 69.3% (77.2% from SG, 60.5% from TH, and 65.6% from K) associated POD with a higher mortality rate (Figure 3).

## 4. Discussion

In this multinational survey, we examined the incidence, risk factors, and outcomes of POD as perceived by anesthesiologists in Southeast and East Asia (K, SG, and TH). To the best of our knowledge, this is the first international survey of the knowledge and perspectives regarding POD among anesthesiologists practicing in these regions.

Postoperative delirium is a common postoperative complication with an incidence as high as 65% in older patients undergoing surgery under anesthesia, and it is associated with increased mortality and morbidity [12]. In general, the respondents in the present study underestimated the incidence of POD in the perioperative period, with approximately half of them estimating an incidence of 11–30%. According to DSM-5 and the Nomenclature Consensus Working Group, POD can occur up to a week after surgery [7,11], although half the respondents in our study were of the opinion that it most commonly occurs within 48 h after surgery. These findings highlight the underestimation of the incidence and time period in which POD can manifest in the postoperative period among the anesthesiologists in the assessed regions.

The anesthesiologists in our survey seemed to focus less on POD as a perioperative problem than on other less common and more catastrophic complications such as myocardial infarction, stroke, and ventilatory problems. Among the cognition-related complications in the perioperative period, intraoperative awareness, which is rare, with an incidence of only 1–2% per 1000 surgical patients, was ranked far more important than POD. This may be attributed to the greater litigious potential of this complication.

POD may manifest as hypoactive, hyperactive, or mixed POD. The hypoactive type of delirium is the most common form, and although it is associated with higher mortality, it may be frequently overlooked [13]. Hypoactive POD is particularly prevalent in intensive care unit settings [14]. Approximately a third of the respondents in our study recognized hypoactive POD as the most common type of delirium.

The American Geriatrics Society (AGS) guidelines recommend preoperative assessment of factors associated with delirium, including age >65 years, chronic cognitive decline or dementia, poor vision or hearing, severe illness, and presence of infection. In addition, a poor functional status, metabolic derangements, polypharmacy, poorly controlled pain, dehydration and poor nutrition, depression, alcohol abuse, sleep disturbances, and a prior history of delirium are common contributing factors. It is unknown if predisposing hereditary vulnerabilities, including sex, genetics, and epigenomic factors, increase the risk of POD in certain patients [10].

While the majority of respondents recognized old age (98.1%) and prior cognitive impairment (90.8%) as risk factors for POD, other risk factors, including prior poor functional status, infection, and metabolic derangements, were not as well recognized. While 44.8% of respondents felt that genetics may be associated with POD, the association remains debatable. Interestingly, 17.3% of respondents mentioned that gut microbiota may be a risk factor for POD, although the concept of the gut–brain axis has only recently gained traction [15].

Mason et al. first reported the association between general anesthesia and POCD, leading to extensive investigations regarding the association between the type of anesthesia and POD or POCD [16]. However, a recent meta-analysis including a heterogenous group of patients showed that, compared with regional anesthesia, properly managed general anesthesia was not associated with an increased risk of POD [17,18,19]. Despite these new findings, nearly all respondents (96.2%) in the present study thought that general anesthesia is associated with POD, and only 30.5% believed that regional anesthesia without sedation was a factor. The same meta-analysis found no evidence of the influence of the type of anesthetic on the incidence of POD, although it found low-certainty evidence that maintenance with propofol-based TIVA could prevent POCD [18]. In contrast, 81.4% and 38.4% of our respondents felt that inhalation agents and propofol contributed to POD, respectively. Ketamine has been proposed as a preventive agent for POD; however, in a well-designed multicenter trial, ketamine had no significant effect on the incidence, severity, or recurrence of delirium [20]. Nevertheless, our respondents gave more importance to ketamine as a contributing factor (74.4%) than to anticholinergics (70.1%).

Benzodiazepine, meperidine, and anticholinergics are strongly associated with POD [10]. Although the majority of respondents (70.1–84.7%) from our survey recognized the contribution of these drugs to POD, 15–35% did not recognize this association. The protective role of dexmedetomidine against POD in adult patients undergoing cardiac and noncardiac surgeries is less recognized, and only 20% of our respondents associated this drug with POD [21,22]. Although neuromuscular blocking agents are not directly associated with POD, 13.7% of respondents related their use to the occurrence of POD.

Considering the little emphasis on the importance of POD and underestimation of its incidence, it is not surprising that only 51.4% of respondents performed a POD test before surgery; the proportion was even lower for patients in PACU and the general ward. This is rather alarming because what is not tested is never diagnosed, and the true incidence of POD will always be underestimated in these regions. The heart and the circulatory and respiratory systems are routinely monitored during anesthesia, whereas monitoring of the brain tends to be neglected in the perioperative period. With an increasingly elderly population presenting for surgery, it is timely to consider routine testing of patients in the perioperative period as part of a brain health initiative [23]. Simple tools such as the Nursing Delirium Screening Scale and the Montreal Cognitive Assessment can be easily incorporated as part of routine perioperative care, particularly for high-risk patients [24,25].

Depth of anesthesia monitoring guided by processed electroencephalography indices could reduce the risk of POD in patients aged ≥60 years who are undergoing noncardiac surgical and non-neurosurgical procedures, and it can also reduce the incidence of POCD at 3 months after surgery in these patients [26]. Although slightly more than three-quarters of our respondents appreciated that these new monitors could reduce the risk of POD, almost a quarter did not know about their usefulness or how to utilize the indices to reduce the patient risk. Again, there was undue emphasis on the prevention of intraoperative awareness, with 88.9% of respondents using these monitors for that purpose and only 52% using it for POD prevention.

There was great variability in the use of depth of anesthesia monitoring among the three countries. Overall, only 13.7% of respondents performed monitoring on a routine basis. The most common reason for limiting routine use was the consumable cost (SG, 50.4%; TH, 53.2%; K, 39.9%). The cost of depth of anesthesia monitoring for patients is USD 140 in K, USD 50 in TH, and USD 28 in SG. Interestingly, even though the cost of single-use devices for depth of anesthesia monitoring is reimbursed by the nationwide public insurance system in K, only a quarter of the respondents surveyed used it on a routine basis. Most likely, this also corresponds to the proportion of respondents (26.6%) who opined that the use of depth of anesthesia monitoring is cost effective. In SG and TH, however, most respondents mentioned that depth of anesthesia monitoring was cost-effective only in high-risk cases; this explained the selective use of these monitors in these countries. Notably, in TH, there is also a scarcity of these resources, which adds to the restricted use of depth of anesthesia monitoring. Ironically, the majority of respondents (83.8%) themselves preferred to undergo surgery with depth of anesthesia monitoring. As evidence builds on the importance of these monitors for POD prevention, whether they should become a part of recommended routine monitors as per the ASA mandatory monitoring standards, particularly in the surveyed regions, is an important question. We believe that our findings from this survey would increase awareness regarding the use of these monitors in the surveyed regions and serve as a starting point for follow-up measures to improve POD detection and prevention.

Newer tools introduced in the surveyed regions include cerebral near-infrared spectroscopy (NIRS), although its efficacy for the prevention of delirium in perioperative or ICU settings is less established [27,28]. Most of the respondents who used NIRS (77%) in the present study used it to prevent stroke, and only 48.4% thought that it could prevent POD. It is evident that the respondents were less experienced in the use of NIRS because only a third of them had used it, and even then, it was only used on an infrequent basis. Furthermore, approximately 10% of respondents mentioned that NIRS is not useful. These misconceptions should be the areas of focus as we define the precise role of NIRS in the prevention of POD and stroke.

POD is strongly associated with long-term NCDs, which can lead to increased morbidity and mortality. Although an overwhelming majority of respondents recognized the association of POD with poor long-term outcomes, surprisingly, only 69.3% recognized that it could contribute to increased mortality. Our results differ from those of surveys involving anesthesiologists from other regions; this suggests that the importance of POD and its impact on mortality is often overlooked in these Asian regions [12,29,30].

The strength of the present survey is the multinational nature and its specific relevance to anesthesiologists practicing in Southeast and East Asia. Nevertheless, there are some limitations. First, the sample population was not standardized across different sites. At K, the participants were residents and faculty members of a teaching hospital in a city. In SG, all responses come from doctors in public hospitals. Only TH included doctors from rural hospitals. Therefore, the results may not accurately reflect the practices in these regions, particularly in rural areas. Second, the questionnaire was designed before the recent introduction of changes by the DSM-5 and Nomenclature Consensus Working Group; therefore, the nomenclature used in the questionnaire could have caused some confusion among the respondents. Third, the current survey concentrated on the delirium of adults. However, POD is also very frequent in children, and the effect continues even after discharge. there is evidence that emergence delirium in children may continue to affect behavior patterns more than 1 week after the surgery [31]. As in adults, there are currently no validated definite treatments for postoperative emergence delirium in children. Therefore, anesthesiologists should focus on the prevention of POD with pharmacological and non-pharmacological interventions, including preemptive screening [9,32].

## 5. Conclusions

The results of the current survey suggest that the incidence and importance of POD are underestimated by anesthesiologists in Southeast and East Asia. Important areas of focus include the recognition, prevention, early detection, and proper management of POD, as these would form the basis for a regional approach to promulgate campaigns such as the “Safe Brain Initiative” to prevent POD-associated adverse outcomes in Southeast and East Asia [1,33].

The first step should focus on raising awareness of POD among medical personnel. It should begin with a routine assessment of the patient’s behavioral, neurological, or developmental status before and after surgery. All surgical patients should be screened before surgery and evaluated again in PACU with well-validated delirium assessment scales such as the Nursing Delirium Screening Scale (NuDESC), the Confusion Assessment. Method for the ICU (CAM-ICU), and the Pediatric Anesthesia of Emergence Delirium (PAED) score [1,9,23]. Routine evaluation can reveal the true incidence of POD, which is often overlooked or underestimated. By recognizing how patients are exposed to POD, preventive measures will become routine. In this way, the POD can be reduced.

## Figures and Tables

**Figure 1 jcm-10-05769-f001:**
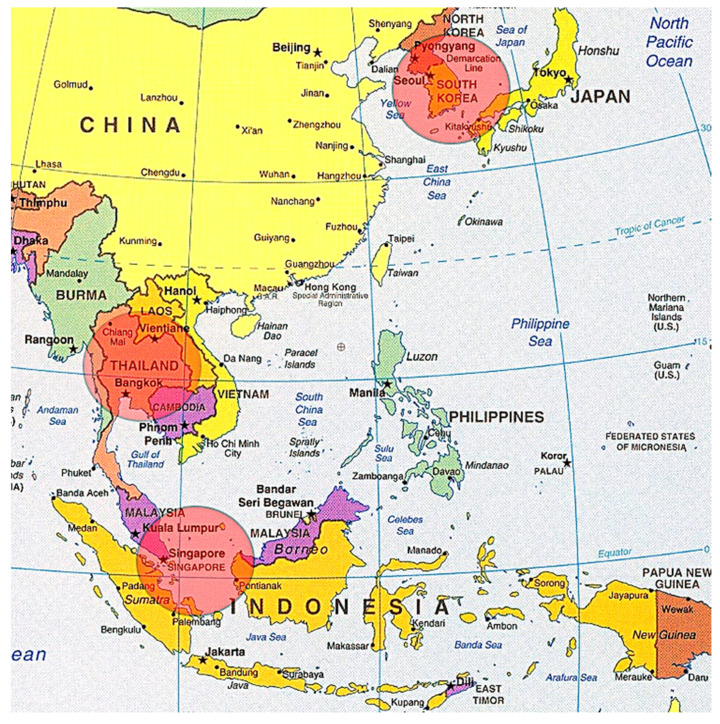
Graphical representation of the survey distribution in the three countries (Thailand, Singapore, and Korea).

**Figure 2 jcm-10-05769-f002:**
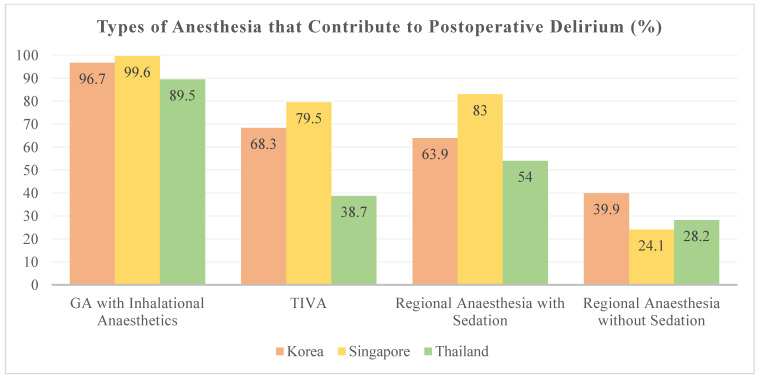
Perceived types of anesthesia associated with postoperative delirium among the survey respondents.

**Figure 3 jcm-10-05769-f003:**
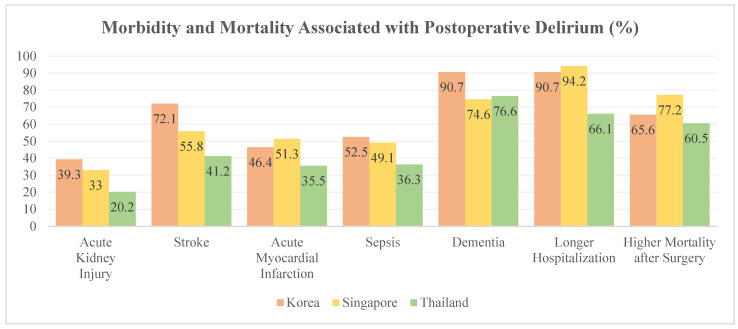
Perceived morbidity and mortality associated with postoperative delirium among the survey respondents.

**Table 1 jcm-10-05769-t001:** Demographics of the survey participants.

Demographics		Number	Proportion
Country	Singapore	224	42.2%
	Thailand	124	23.3%
	Korea	183	34.5%
Age (years)	Less than 30	114	21.5%
	31 to 40	215	40.5%
	41 to 50	115	21.6%
	More than 50	87	16.4%
Sex	Male	235	44.3%
	Female	296	55.7%
Main Location of Practice	University Hospital	335	63.1%
	City Hospital	154	29.0%
	Regional or Rural Hospital	42	7.9%
Years of Practice	Less than 5	184	34.7%
	5 to 10	145	27.3%
	11 to 20	118	22.2%
	More than 20	84	15.8%

**Table 2 jcm-10-05769-t002:** Respondents’ rankings for the importance of postoperative complications and cognitive states.

Survey Questions		Korea	Singapore	Thailand
Rank in order of importance the following postoperative complications (1—most important; 8—least important)	Myocardial infarction	2	2	1
	Stroke	3	1	2
	Hypoventilation & CO_2_ Retention	1	3	3
	Hypothermia	5	5	4
	Postoperative Delirium	6	4	5
	Pain (Visual Analog Scale score > 3)	4	7	7
	Postoperative Nausea and Vomiting	8	6	6
	Surgical Site Infection	7	8	8
Rank in order of importance the following cognitive states (1—most important; 4—least important)	Intraoperative awareness	1	1	1
	Postoperative cognitive dysfunction	2	2	3
	Postoperative delirium	3	3	4
	Emergence agitation	4	4	2

**Table 3 jcm-10-05769-t003:** Perceived risk factors for developing postoperative delirium among the survey respondents.

Survey Question		Number	Proportion
What are the risk factors for developing postoperative delirium? Tick all that apply.	Age > 65 years	521	98.1%
	Only primary school education	212	39.9%
	Poor nutritional status	404	76.1%
	Frailty	306	57.6%
	Anemia	299	56.3%
	Diabetes	258	48.6%
	Renal insufficiency	280	52.7%
	Cognitive impairment	482	90.8%
	Genetics	238	44.8%
	Gut microbiome	92	17.3%
	Others	22	4.1%

## Data Availability

Survey questions and responses are publicly available on Mendeley Data: Hyungmook Lee, Jeongmin Kim, Ki-young Lee, Tong J. Gan, Varinee Lekprasert, Prok Laosuwan, Sophia Tsong Huey Chew, Edwin Seet, Vera Lim, and Lian Kah Ti (2021), “Survey questions and responses of Awareness and Perspectives Among Asian Anesthesiologists on Postoperative Delirium: A Multinational Survey”, Mendeley Data, V2, doi: 10.17632/rc3jkwhf7f.2

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
