# Peer review of "Awareness and Perspectives among Asian Anesthesiologists on Postoperative Delirium: A Multinational Survey"

_jcm, 2021, doi:10.3390/jcm10245769_

Round 1
Reviewer 1 Report
Peer review:
Awareness and Perspectives Among Asian Anesthesiologists on Postoperative Delirium: A Multinational Survey
Lee et al. present the results of a multinational survey assessing the current knowledge on postoperative delirium among anesthesiologists in Singapore, Thailand and South Korea. Interestingly, the relevance of postoperative delirium seems still to be underestimated and the current survey provides further insights in how to address this with future education programs. Overall, the topic is of general interest, the manuscript is well written, and the results are adequately presented. I therefore recommend acceptance after consideration of one minor point as follows.
The last paragraph of the introduction includes the explanation on how postoperative delirium is defined. I think the definition used in the current study doesn’t add much to the justification of the current study, which the introduction should be all about. I therefore suggest moving this part to the methods section. The aim of the current study (last sentence) should be presented with a separate paragraph and stay at the end of the introduction.
Reviewer 2 Report
Thank you for submitting the manuscript. I read your paper with great interest, which deals with an interesting and highly debated topic. Your manuscript is very smooth and clear. However, I have some requests to make of you in order to improve the quality of the manuscript.1. In the introduction you talk about delusion, with emphasis on incidence and consequences. This is admirable, but leave out a population where delirium is unfortunately very frequent: children. I ask you to also deal with delirium in the pediatric field, both in the introduction and in the discussion. I would like to point out some references to deepen the subject: doi: 10.1097 / MD.0000000000005566. doi: 10.1097/ACO.0000000000000076. doi: 10.2147/DDDT.S163828. doi: 10.23736/S2724-5276.21.06330-8. doi: 10.1093/bja/aew477.
2 Since this is a survey, it is necessary to provide the total number of subjects to whom the questionnaire was administered, divided by country, and the response rate, both global and by country
3 How did you choose the anesthesiologists to administer the questionnaire to? How did you choose the hospitals? Why did you exclude some of them?
4 The percentage of anesthetists who answered what is the total number of anesthetists working in a single country and in the area you have indicated as the union of the three countries?
5 Finally, I would like you to insert a paragraph on "perspectives" indicating what are the guidelines for training and health policies that this survey should suggest.
I hope my comments are useful to you
Kind Regards
Round 2
Reviewer 2 Report
Thank you for submitting the new version of the manuscript. I can be satisfied with the work you have done, having answered all my questions. The manuscript is indeed very valuable now.
Kind Regards